# Pharmacotherapies Targeting GABA-Glutamate Neurotransmission for Treatment-Resistant Depression

**DOI:** 10.3390/ph16111572

**Published:** 2023-11-07

**Authors:** Courtney M. Vecera, Alan C. Courtes, Gregory Jones, Jair C. Soares, Rodrigo Machado-Vieira

**Affiliations:** 1Department of Psychiatry and Behavioral Sciences, University of Texas Health Science Center, Houston, TX 77054, USA; 2John S. Dunn Behavioral Sciences Center at UTHealth Houston, 5615 H.Mark Crosswell Jr St, Houston, TX 77021, USA

**Keywords:** antidepressants, depression, treatment-resistant depression, treatment, targets, GABA, glutamate, pharmacotherapies

## Abstract

Treatment-resistant depression (TRD) is a term used to describe a particular type of major depressive disorder (MDD). There is no consensus about what defines TRD, with various studies describing between 1 and 4 failures of antidepressant therapies, with or without electroconvulsive therapy (ECT). That is why TRD is such a growing concern among clinicians and researchers, and it explains the necessity for investigating novel therapeutic targets beyond conventional monoamine pathways. An imbalance between two primary central nervous system (CNS) neurotransmitters, *L*-glutamate and γ-aminobutyric acid (GABA), has emerged as having a key role in the pathophysiology of TRD. In this review, we provide an evaluation and comprehensive review of investigational antidepressants targeting these two systems, accessing their levels of available evidence, mechanisms of action, and safety profiles. N-methyl-D-aspartate (NMDA) receptor antagonism has shown the most promise amongst the glutamatergic targets, with ketamine and esketamine (Spravato) robustly generating responses across trials. Two specific NMDA-glycine site modulators, D-cycloserine (DCS) and apimostinel, have also generated promising initial safety and efficacy profiles, warranting further investigation. Combination dextromethorphan-bupropion (AXS-05/Auvelity) displays a unique mechanism of action and demonstrated positive results in particular applicability in subpopulations with cognitive dysfunction. Currently, the most promising GABA modulators appear to be synthetic neurosteroid analogs with positive GABA_A_ receptor modulation (such as brexanolone). Overall, advances in the last decade provide exciting perspectives for those who do not improve with conventional therapies. Of the compounds reviewed here, three are approved by the Food and Drug Administration (FDA): esketamine (Spravato) for TRD, Auvelity (dextromethorphan-bupropion) for major depressive disorder (MDD), and brexanolone (Zulresso) for post-partum depression (PPD). Notably, some concerns have arisen with esketamine and brexanolone, which will be detailed in this study.

## 1. Introduction: The Problem of TRD

Depression is an increasingly prevalent and debilitating psychiatric disorder with a heterogeneous symptomatic picture and complex neurobiological basis. In the United States, depression is the leading cause of disability and suicide, affecting over 17.3 million adults [1]. Major depressive disorder (MDD) costs Americans approximately $210.5 billion annually, whereas the global economic burden of depression and anxiety is estimated to be one trillion USD annually and rising [1,2]. Approximately one-third of depressed patients fail to remit, even after four adequate therapeutic trials [3]. Diminishing returns demonstrated in the Sequenced Treatment Alternatives to Relieve Depression (STAR*D) trial underscore the need for novel treatment avenues targeting the pathophysiological source of treatment resistance [4]. Disappointingly, no clear consensus exists for the definition of TRD. Lack of response or remission after two adequate trials with standard antidepressant therapies appears to be the modal definition. However, only 17% of studies implement these criteria [5]. For the purposes of this review, TRD will be defined as such.

Overall, Treatment Resistant Depression (TRD) is associated with several comorbid features, including prolonged mental and physical dysfunction, increased healthcare spending, worse clinical outcomes, and a higher risk of suicide [4,6,7,8,9]. Well-established clinical correlates of TRD include persistent anhedonia and anxiety, the presence of one or more medical and/or psychiatric comorbidities, as well as duration and frequency of depressive episodes [10]. 

Most of the available FDA-approved pharmacological treatments for MDD target conventional monoamine pathways (i.e., serotonin, dopamine, and norepinephrine), which make up less than ten percent of total central nervous system (CNS) activity [11]. Thus, there is an urgent need for new, improved antidepressant therapies targeting a broader range of neurotransmission. The present study provides an overview of novel, rapid-acting antidepressants with potential efficacy in TRD based on a physiological balance between the brain’s two primary neurotransmitters, glutamate and GABA. 

## 2. Glutamate and GABA Dysfunction in Depression

The function of the CNS fundamentally relies on a delicate physiological balance between glutamatergic and GABAergic systems. With more than 90% of CNS neurons acting through these pathways, the excitatory activity must be well-regulated by an inhibitory component [12]. Glutamate hyperactivation associated with impaired GABA inhibition creates detrimental neural physiology, changing gene expression, cellular morphology, and signaling activity. Receptors that are able to receive and process the signals from glutamate or GABA are present on all cells in the brain, including neurons and glia [13]. Abnormalities in volume, activity, and connectivity in cortico-limbic networks have been consistently linked to depressive pathophysiology [14,15]. Glutamatergic and GABAergic dysfunction in the prefrontal cortex (PFC) and anterior cingulate cortex (ACC) has been extensively implicated in both MDD and TRD [16,17,18,19,20,21,22,23,24,25]. The default mode network (DMN), one of the CNS’s major communication networks has also been implicated in this dynamic. Specifically, cortical GABAergic disinhibition in depressive disorders co-occurs with increases in glutamatergic gene expression in the DMN [24,26,27,28,29,30,31,32].

### 2.1. Glutamatergic Abnormalities

Glutamate activity plays a key role in learning and memory, synaptic plasticity, and overall behavior [33]. Moreover, glutamatergic transmission appears to be a key mediator of mood, cognition, perception, and emotions associated with TRD [15]. A meta-analysis of 1H-MRS studies demonstrated that decreased Glx levels with absolute values in the prefrontal cortex were correlated with treatment severity (i.e., number of failed antidepressant treatments), indicating that the severity of glutamatergic dysregulation could be related to the severity of illness [34]. Subjects with depression have been shown to display a variety of glutamatergic abnormalities, including reduced glial density, decreased expression of the glutamate reuptake transporters EAAT1 and EAAT2 and decreased enzymatic conversion from glutamate to glutamine [33].

One proposed neurobiological mechanism underlying TRD is related to the toxic effect of extrasynaptic glutamate receptor hyperactivation. As extracellular glutamate release outpaces clearance by glial cells, and the ensuing inflammation and neurodegeneration likely contribute to the acute volume reductions and other cytoarchitectural abnormalities detected in depressed patients’ brains [35]. To counteract these effects, glial cells are responsible for glutamate reuptake and facilitate the glial–astrocytic conversion of glutamate to glutamine, which limits excitotoxicity and provides necessary precursors for GABA synthesis [36]. Glutamine, the most abundant amino acid in the CNS by an order of magnitude, also plays a vital role in cellular buffering, transcription/translation, mitochondrial functioning, and other vital processes [33].

Metabotropic glutamate receptors (mGluRs) are highly expressed in brain regions central to the pathophysiology of TRD. These receptors also influence local GABA and glutamate activity; mGluR_5_ interacts with glutamatergic and GABAergic neurons throughout the interconnected circuitry of the PFC, hippocampus, and amygdala. They control processes such as learning, memory acquisition, fear extinction, and synaptic plasticity [37]. In this capacity, learning performance and response depend on the frequency of mGluR_5_ expression in the hippocampus [38,39]. The mGluR_5_ receptor also mediates stress resilience via post-synaptic mGluR_5_ activation on the nucleus accumbens [40,41]. This process may ultimately facilitate hippocampal neurogenesis and normalize hypothalamic–pituitary–adrenal (HPA) axis activity, two processes that have been repeatedly implicated in recovery from depression [42,43].

Ionotropic glutamate receptors also play a vital role in synaptic plasticity [44]. In depressed patients, synaptic plasticity pathways have been shown to be disrupted in the PFC and hippocampi, correlated with α-amino-3-hydroxy-5-methyl-4-isoxazolepropionic acid (AMPA) and NMDA receptor abnormalities [45,46]. The possibility of development of antidepressant medications targeting the NMDA complex was suggested decades ago [47]. Regardless of causal direction, the reciprocal and downstream effects of this dysfunction substantially contribute to neurodegeneration within the PFC and hippocampus. These areas are highly implicated in the adverse cognitive and affective features of depression, especially rumination and anhedonia [3,48].

Brain-derived neurotrophic factor (BDNF) levels, a primary driver of neuroplasticity and glutamate modulation, are reduced in postmortem hippocampal and PFC samples of patients with MDD. It is suggested that alterations in BDNF activity may contribute to the acute regional volumetric decreases associated with MDD/TRD [3]. Moreover, multiple SNPs in BDNF-associated regions have been associated with treatment response to ketamine and selective serotonin reuptake inhibitors (SSRIs) in some patient populations. BDNF knockout mice also show reduced responsiveness to such therapies [3,49]. BDNF and its cellular target (tyrosine kinase receptor B (TrkB)) potently regulate neuronal survival and growth through several downstream effectors: bcl-2, mammalian target of rapamycin (mTOR), glycogen synthase kinase-3B (GSK-3B), and phosphatidylinositol 3-kinase (PI3-kinase)/Akt [50].

Increasing evidence suggests that BDNF-TrkB signaling underlies a substantial portion of both affective pathophysiology and treatment response across therapeutic approaches [51,52]. Importantly, the rate of activity-dependent BDNF release appears to distinguish rapid-acting agents discussed in this review from their monoaminergic counterparts, whose delayed treatment response coincides with an indirect increase in BDNF secretion weeks after initiation [53]. It has recently been demonstrated that conventional antidepressants such as fluoxetine and imipramine, as well as the rapid-acting glutamatergic agent ketamine, directly bind to TrkB and allosterically potentiate BDNF signaling [52,54]. This may suggest a final common pathway for many antidepressant modalities—amplification of endogenous glutamate/BDNF signaling via TrkB. Notably, multiple serotonergic hallucinogens have also demonstrated preliminary efficacy in TRD [55,56] and impart downstream glutamatergic effects (as well as robust spinogenesis and dendritogenesis), which are dependent on intact TrkB signaling [57,58]. Lysergic acid diethylamide (LSD) and psilocin have recently been found to directly bind to TrkB with affinities 1000-fold higher than those for conventional antidepressants and ketamine. However, 2R,6R-hydroxynorketamine (R,R-HNK), an active metabolite of ketamine with negligible affinity for NMDA receptors, was found to displace LSD from TrkB at high nanomolar concentrations, suggesting relatively comparable potency [54]. Despite repeated and robust demonstration of antidepressant effects across animal models with R,R-HNK, higher plasma levels appear to confer less improvement in clinical studies of depression and suicidal ideation [59,60]. Surprisingly, in patients with TRD, higher R,R-HNK levels have been shown to correlate with increased encephalographic gamma power—a putative measure of cortical disinhibition/excitation (i.e., decreased activity in GABAergic interneurons and increased activation of fast ionotropic glutamate receptors on pyramidal neurons) [59]. Thus, it remains unclear as to whether the cortical “glutamate surge” and/or resultant TrkB signaling are sine qua non to antidepressant efficacy for this class. The first phase I trial with R,R-HNK is currently underway (NCT04711005), which may hopefully add some clarity. Further caveats to this hypothesis exist as well, and it is important to place any BDNF-related findings within the context of population heterogeneity and neuro-regional specificity [50,53].

### 2.2. GABAergic Abnormalities

Strong evidence supports a key role for GABA in MDD, showing structural and functional GABAergic system deficits throughout the central and peripheral nervous systems [61]. Compared to healthy controls, depressed patients consistently display net reductions in cortical GABA concentrations and neuronal density, as well as decreased enzymatic synthesis in the periphery and cerebrospinal fluid (CSF) [61,62]. In women with TRD/treatment-resistant postpartum depression, depressive severity is linked to GABA concentrations in the DMN [26,27]. In the CNS, reductions in ACC GABA levels appear to correlate with increased anhedonia and treatment resistance [19,23]. In contrast, higher baseline ACC activity is associated with improved outcomes [22]. In TRD specifically, rostral ACC function may mediate the balance between negative rumination and constructive self-evaluation, both facilitated by the DMN [17,22], suggesting that ACC dysfunction plays a role in excessive negative rumination. In both MDD and TRD, region-specific normalization of GABA concentrations in the occipital cortex (OCC), ACC, and DMN have repeatedly been observed in response to all successful antidepressant therapies [19,22,23,63].

Likewise, reductions in both GABA_A_ and GABA_B_ receptor-mediated inhibition in MDD have been demonstrated across genomic [64], postmortem [65], and neuroimaging studies [28,48]. Levinson and colleagues’ transcranial magnetic stimulation (TMS) study demonstrated substantial deficits in GABA_A_ inhibitory signaling in patients with treatment-resistant depression, but not in those with MDD or euthymic remitters with a history of MDD [28]. These results suggest that neurophysiological deficits of the GABA_B_ receptor are more broadly related to depressive pathophysiology and symptoms, whereas GABA_A_ receptor deficits are more selectively associated with illness severity and treatment response [28]. Overall, MDD and TRD seem to be biologically different. GABAergic deficits involve the concomitant presence of neurophysiological, neuroendocrine, cognitive, and behavioral findings [32,66,67], which may be reversed with targeted therapeutics.

## 3. Methods: Clinical Studies of Investigational Compounds Targeting Glutamate and GABA

The aim of this study is to present most relevant scientific research regarding the several investigational compounds targeting the glutamatergic and/or GABAergic systems as their primary mechanism of antidepressant action that have been tested in clinical trials up to September 2023. All proceeding compounds demonstrate evidence of antidepressant efficacy in at least one successful clinical trial, though not all have progressed through the final stages of drug development. (For an overview of investigational antidepressants targeting GABA/glutamate, including level of evidence for antidepressant efficacy, see Table 1 for GABAergic compounds and Table 2 for glutamatergic compounds.) On Table 3, we provide a guide on how we defined the evidence levels of each compound included on Table 1 and Table 2. We have briefly mentioned several psychedelic compounds above, two of which (ayahuasca and psilocybin) have demonstrated preliminary efficacy in TRD [55,56]. As the extent of their direct glutamate/GABA modulation in relation to antidepressant efficacy is unclear at this juncture, they have not been formally included in our review.

## 4. Clinical Studies with Glutamate Modulators in TRD

Anhedonia and anxiety represent clinical markers of treatment resistance in a significant proportion of people with depression; therefore, assuaging these symptoms is a key component of several of the following investigational drugs. Glutamatergic modulation in depression is, to a degree, already a successful mechanism of many conventional antidepressants (cAD), refining neurotransmission and receptor expression via direct and indirect actions with chronic use [68,69].

Diverse NMDA receptor antagonism has demonstrated rapid antidepressant efficacy in preclinical depression paradigms [24,29,70,71] and in some clinical trials [72,73]. Notably however, whereas glutamatergic neurotransmission appears critical to the therapeutic efficacy of these compounds, whether direct NMDAR antagonism is primarily responsible for the ultimate effects is very much unsettled [74]. Paradigms such as GABAergic intraneuronal NR2B blockade leading to an extracellular glutamate surge and NMDA-mediated AMPAR potentiation have been proposed. However, precise mechanisms have remained obscured amidst the (at times conflicting) agonist/antagonist, metabolite, and knockout studies addressing this issue. For now, the therapeutic primacy of NMDAR inhibition and resultant AMPAR potentiation appears a temporary frontrunner in the dynamic glutamatergic antidepressant landscape [75,76]. Phase I trials using the AMPAR antagonist perampanel (Fycompa) to potentially negate the antidepressant effects of ketamine are underway (NCT03367533, NCT03973268), which may add more clarity to the situation.

The ensuing sections will analyze the highest level of published data from investigational antidepressant agents that agonize, antagonize, or otherwise affect glutamatergic neurotransmission, in the context of TRD pharmacotherapy.

### 4.1. Ketamine and Similar Compounds

Ketamine, arguably the most well studied rapid-acting antidepressant and anti-suicidal agent of the last two decades, is a derivative of phencyclidine (PCP). First approved by the US Food and Drug Administration (FDA) in 1970 as a dissociative anesthetic and pain management agent, ketamine’s antidepressant efficacy at sub-anesthetic doses was not discovered until thirty years later [72]. Various hypotheses of ketamine’s rapid AD effects implicate cortical NMDAR-AMPAR throughput, BDNF-TRKb-mTOR enhancement, monoamine enhancement, opioid receptor signaling, immunomodulation, and others as the mediators of downstream antidepressant effects. Such complex effects seem to ultimately result in increased neurogenesis and synaptogenesis, improved neuroplasticity, and normalization of stress-related neurodegeneration [77,78].

However, several lines of evidence must be considered. First, contrary to preclinical observations, a recent randomized controlled trial (RCT) in humans demonstrated prolonged antidepressant responses to ketamine with coadministration of rapamycin (mTOR inhibitor) [79]. While these findings necessitate further validation, they present several important lines of clinical inquiry. Second, another study demonstrated that the induction of antidepressant behavioral effects occurs in vivo independently of (and before) actual spine formation. However, glutamate-induced synaptogenesis in their study played a critical role in sustaining the effects over the subsequent week [80].

Regarding other receptors, a complex relationship exists between opioid receptor signaling and glutamate transmission. Both systems appear essential, though not independently sufficient, to induce ketamine’s AD action [81,82,83]. Some (but not all) studies have demonstrated attenuation of ketamine’s AD efficacy with coadministration of naltrexone, a non-selective opioid receptor antagonist, again necessitating further replication [84,85]. Lastly, contrasting the deleterious effects seen in heathy adults or chronic ketamine addiction, repeated acute phase administration of IV ketamine appears to reliably ameliorate cognitive impairment in TRD patients [86]. Improvements in working memory might be predictive of anti-suicidal responses in this population as well [87]. Taken together, these results suggest important translational gaps between animals and humans, necessitating a greater emphasis on patient biomarkers and phenotyping in clinical trials.

In that regard, lower baseline gamma power (reflective of GABA and glutamate functional balance) and subsequent elevation after ketamine administration have been shown to predict better antidepressant responses in TRD [88], though further validation is needed. Ketamine administration in depressed patients also appears to reconfigure functional connectivity within the DMN. Notably, normalization of insular DMN hyperactivity in TRD patients appears to coincide with peak antidepressant responses (48 h), subsequently reversing at 10 days post-infusion (when effects on mood are typically lost) [89]. Elevation in peripheral BDNF appears to be one of, if not the only, robustly validated blood-based marker for ketamine response. Importantly, BDNF has displayed adequate central–peripheral correlation, being reflective of CNS glutamatergic activity [90].

In terms of clinical trials, Berman and colleagues’ randomized, double-blind, placebo-controlled trial (RDBPCT) was the first clinical report of ketamine’s rapid and robust antidepressant effects [72], marking a watershed in the development of antidepressant therapies. Comparable results have been replicated in several similarly designed RDBPCTs examining single administration of sub-anesthetic dose ketamine in patients with MDD and TRD [73,91,92,93]. Meta-analyses consistently support these findings [91,92,94], establishing ketamine’s AD efficacy in oral and intranasal administration, repeated dosing, an augmenting agent, and patients with treatment-resistant bipolar depression or suicidal ideation [91,94,95,96].

Across adjunctive trials in TRD, a single dose of ketamine can alleviate depressive symptoms for up to seven days, with AD effects peaking 24 h after administration [91]. In MDD, these effects are maintained for two to three weeks with repeated infusions [91]. In terms of sustained effects, two RCTs have shown prolonged (4–6 weeks) antidepressant effects using combined intravenous (IV) (0.5 mg/kg) and oral (50 mg/day) ketamine augmentation in MDD/TRD populations [95,96]. However, at the onset of both trials, patients were newly initiated on SSRIs, thus obfuscating the timing of their improvement. Combining ketamine with other NMDA allosteric modulators such as D-cycloserine (DCS) or riluzole may also prolong anti-suicidal effects for over 6 weeks in TRD patients [97]. Of note, multiple preclinical and clinical studies suggest that ketamine’s anti-suicidal effects are at least partially disparate from its AD action [98].

Although results are promising, limitations have emerged across RCTs, most notably the inability to effectively blind subjects and raters to the drug [99]. Additionally, many studies are hindered by homogenous samples, short durations, and limited real-world feasibility [98,100]. Data from recreational ketamine users show that long-term ketamine use can lead to renal and bladder toxicity, cognitive impairment, dependence, and induced suicidal ideation during withdrawal [101,102]. This has fueled the search for novel agents similar to ketamine but without the undesirable side effects [103]. One such investigational compound is the *S* (+) enantiomer of racemic ketamine: (*S*)–ketamine hydrochloride or esketamine.

Esketamine has nearly fourfold greater affinity for the NMDAR than (R)-ketamine but interestingly induces fewer and weaker psychotomimetic and dissociative effects [103]. Esketamine’s putative mechanism of action likely involves a similar, transient surge in glutamatergic transmission, accompanied by increased synaptogenesis and an influx in neurotrophic factor activity [46,77,97]. In 2019, esketamine nasal spray (marketed as Spravato) was approved as an adjunctive AD treatment for TRD in the United States and the European Union, becoming the first novel-acting antidepressant approved in decades [104,105]. In contrast, the National Health System (NHS) of Great Britain contemporarily rejected the drug, citing unconfirmed benefits, inflated price, and severe adverse effects in its decision [106]. Independent groups conducting cost-effectiveness analyses have consistently determined that esketamine/Spravato exceeds standard cost-effectiveness thresholds and is currently too expensive for widespread or long-term use in the United States [107,108].

To date, esketamine’s AD efficacy has been tested in five phase III RDBPCTs and one phase III open-label clinical trial as an augmentation strategy, mostly to new oral antidepressants (NOA). Results from two 28-day phase III RCTs support multi-dose, adjunctive esketamine’s tolerability and AD efficacy in adult and geriatric samples [109,110]. Based on positive phase II results, Popova and colleagues hypothesized a far greater effect size than what manifested from their trial, possibly related to unsuccessfully blinding esketamine’s subjective effects, especially in comparison to inactive placebo [110]. Fedgchin’s group conducted an analogous study, which did not support adjunctive esketamine’s AD efficacy for TRD [111]. Most recently, Fu and colleagues demonstrated moderately significant AD effects with adjunct intranasal (IN) esketamine but failed to demonstrate an effect for suicidality, the trial’s other primary outcome. In fact, with seven suicide attempts and one completion by a patient who had received esketamine just three days prior, Fu and colleagues’ findings may draw a troubling parallel to SSRI black box warnings [104].

A recent, longer (4 months) clinical trial by Daly and colleagues demonstrated a significant dose–response curve associated with esketamine augmentation [112]. The results suggest that prolonged treatment may help delay relapse time among TRD patients who respond well to short-term adjunctive esketamine. Long-term esketamine use in Daly and colleagues’ study was also linked to short-term, adverse effects, including increases in blood pressure, dissociative experiences, and incidents of worsening depression when compared with placebo. Findings regarding both safety and efficacy in the trial are of limited applicability due to a high dropout rate [112]. Wajs and colleagues conducted the longest phase III trial of intermittent esketamine augmentation to NOA with the median open-label esketamine exposure of just under 23 weeks [113]. A very recent open-label, long-term study accessed the safety and efficacy of esketamine use in patients with TRD [114]. The results were positive, showing that the same improvement of depressive symptoms at the first 4 weeks of exposure to the drug were sustained during the 2-year mark at the end of the study, with a very small percentage of patients discontinuing the study because of worsening depression (0.6%) or suicide ideation (0.2%). A successful suicide attempt was reported among patients receiving esketamine. Although interpreted by the investigator as unrelated to esketamine, the suicide completer had no history of suicidal behavior or intent [113]. No deaths were reported in the placebo group. Wajs and colleagues suggested that the results provide moderate support for long-term, adjunctive esketamine in TRD [113].

As detailed above, esketamine-associated adverse events across trials have raised significant concern, particularly with regard to inducing suicidality [115]. Janssen-sponsored investigators have also weighed in on this issue [116]. Currently, patients receiving prescription esketamine must adhere to Risk Evaluation and Mitigation Strategy (REMS), a restrictive medical distribution and observation program [117]. Owing to these drawbacks, intensive efforts to develop safer glutamatergic antidepressants with rapid onset, robust efficacy, and sustained symptom reduction have commenced globally [103].

Notably, a recent meta-analysis has indicated significantly greater response (RR = 3.01 vs. RR = 1.38) and remission rates (RR = 3.70 vs. RR = 1.47), as well as lower dropouts (RR = 0.76 vs. RR = 1.37), favoring intravenous ketamine over intranasal esketamine [118]. The only head-to-head comparison to our knowledge (*n* = 63) found that when both compounds are delivered intravenously, they exert similar rapid antidepressant effects in TRD patients but trend towards favoring racemic ketamine at 7 days (*p* = 0.08) [119]. There is an ongoing phase IV trial accessing the clinical response of esketamine monotherapy in patients with TRD that compares two different doses and placebo accessing the MADRS from day 1 to week 4 [120]. Given limited resource availability and acuity for TRD patients, future effectiveness studies comparing intranasal esketamine to IV racemic ketamine would seem warranted. Blinding can be accomplished by simultaneous IV/IN administration.

Key enantiomeric differences beyond route-dependent absorption might also contribute to these findings.

Arketamine (R(-)-ketamine, PCN-101) has demonstrated results in its first open-label study delivering single IV infusion to seven TRD patients [121]. Another open-label study (Crossover RCT) demonstrated mixed results, where a single infusion was not superior to placebo in improving depressive symptoms on TRD patients. [122]. A larger phase II RCT (*n* = 102) in TRD has recently been completed, with negative results. The trial did not meet both primary and secondary endpoints, with no statistical differences between the MADRS results after 24 h of administration on all three arms [123]. Notably, preclinical studies suggest that this enantiomer may have important downstream signaling differences from esketamine, conferring more prolonged responses with fewer adverse effects compared to either racemic or esketamine [124]. Further RCTs (possibly including adjunctive use) may be warranted, given mixed initial signals.

Nitrous oxide (N_2_O), known widely as “laughing gas”, is primarily a non-competitive NMDA receptor inhibitor and dissociative anesthetic. N_2_O is one of the World Health Organization’s essential medicines, and its clinical administration has been considered safe for more than 150 years [125]. Nagele and colleagues’ proof-of-concept RCT tested the antidepressant effects a single session of nitrous oxide inhalation in TRD subjects and reported substantial, rapidly onset AD effects with fewer adverse outcomes than anticipated [99,126]. A very similar and recent study had some mixed findings on a larger sample (N = 44), showing improvements in MDD symptoms during the first 24 h after inhalation but no significant differences from placebo after one and two weeks [127]. Concerns regarding toxicity and abuse risk, which mostly occur with chronic, high-dose recreational usage, have somewhat obscured promising initial findings [128]. Notwithstanding positive results from a triple crossover RCT comparing different concentrations in MDD suggest feasibility. Several trials with N_2_O are currently enrolling for depression (MDD and bipolar), suicidal ideation, post-traumatic stress disorder, and others [129].

d-Methadone (dextromethadone/REL-1017) is the d-stereoisomer of methadone. Methadone itself has repeatedly demonstrated efficacy treating depression among opioid use disorder patients [130]. The d-stereoisomer likewise acts as an NMDA receptor antagonist, binding at the same synaptic site as ketamine and dizocilpine (MK-801) [131,132]. Human and animal trials have not linked d-methadone to any reinforcement, respiratory depression, or other adverse effects seen with opioids or ketamine [131]. However, some have questioned whether it is truly free of abuse potential [133]. A phase IIa RDBPCT evaluated the AD efficacy of one loading dose followed by daily oral d-methadone as an add-on to TRD patients’ existing therapies [134]. Compared to placebo, subjects in both d-methadone treatment groups began showing meaningful improvements across depressive outcome measures by day four, which persisted for one week following the subjects’ last dosage [134]. Results from a sub-analysis made from the phase II study accessing subjective cognitive measures from both MADRS and SDQ showed improvement in cognitive functions in the groups that received the drug compared to placebo in 14 days of evaluation [135]. Negative top-line results from the first phase III TRD study RELIANCE III were recently released. Participants (*n* = 232) were given d-methadone monotherapy for 28 days, showing a 14.8-point reduction in the MADRS on day 28. However, placebo arm demonstrated a 13.9-point reduction, with some sites seeing placebo dramatically outperform the active treatment [136]. The other phase III study (RELIANCE I) had negative results, when the drug was tested as monotherapy for treatment of MDD [137]. Another phase III study (RELIANCE II) assessing its adjunctive use in MDD is underway [134,138].

MIJ821 (Novartis) is another novel NMDA receptor antagonist with minimal psychomimetic effects. The first phase II study in TRD patients recently completed. The 6-arm, dose-finding RCT (*n* = 70) included placebo and IV ketamine comparator arms. Both low and high doses separated from placebo at 24 and 48 h but not at 6 weeks (preliminary results were reported within an 80% confidence interval) [139]. A larger phase II, 7-arm RCT in TRD is ongoing (NCT05454410) is ongoing as well as another in MDD with active suicidal ideation (NCT04722666) with expected results towards the end of 2023 and 2024.

Decoglurant (Roche), an mGlu2/3 antagonist receptor, was tested in a large clinical trial (*n* = 370), where the primary objective was to access antidepressive and procongnitive of the compound versus placebo as an adjunctive therapy to SSRI/SNRI therapy in individuals with TRD [140]. At 6 weeks, decreases in MADRS scores were noted from baseline, but without reaching statistical significance when compared to the placebo group, failing the primary endpoint, and, for the secondary endpoints, mood and functioning and cognitive impairment, it has also failed in separate from the placebo group.

TS-161 (Taisho Pharmaceutical) is also an mGlu2/3 antagonist receptor, like the previous compound on our review, but this one was only tested in a preclinical–clinical phase I trial [141]. This study found that the compound is safe and well-tolerated in humans, but with the need of further clinical development to access the treatment of MDD. At the current moment, there is an ongoing phase II crossover RCT at NIMH (NCT04821271), where patients with MDD will take TS-161 for 3 weeks and placebo for 3 weeks, with the MADRS score at day 21 as the primary endpoint, with an estimated completion date in 2024.

Other exploratory AD compounds that exemplify NMDAR antagonism without ketamine’s associated psychotomimetic and dissociative properties include memantine [131], lanicemine [142], and CERC 301 [143]. In all reports, initial efficacy was either transient or inconsistent in larger studies [97,103,144,145,146].

### 4.2. Dextromethorphan and Similar Compounds

Dextromethorphan (DM) is a nonselective, uncompetitive NMDAR antagonist, therapeutically implicated in various neuropsychiatric disorders [147]. The unconfirmed mechanism of AD action may rely on σ_1_ (“opioid-like”) receptor activation, monoaminergic neurotransmitter reuptake inhibition, and nicotinic acetylcholine (nACh) receptor antagonism, leading to downstream AMPAR activation [147,148,149]. Due to its short half-life, DM lacks AD efficacy as a monotherapy, but in conjunction with quinidine, a CYP2D6 enzyme inhibitor that limits DM metabolism, it produces more lasting AD effects [148,149]. A study group found negative results when combining dextromethorphan to valproic acid in order to improve depressive symptoms in patients with bipolar disorder [150].

AVP-923/Nuedexta consists of a combination of the NMDA antagonist/sigma1 receptor agonist dextromethorphan hydrobromide (DM) and the cytochrome P450 2D6 (CYP2D6) enzyme inhibitor quinidine sulfate [151]. It exhibited high response rates without the emergence of suicidal ideation, dissociation, or psychotomimetic effects in an early phase II clinical trial with TRD patients [152]. However, the trial’s high dropout rate and lack of a comparator group limited the generalizability of the findings. Nuedexta is FDA-approved for pseudobulbar affect treatment, and alternative formulations designed to increase DM’s bioavailability without the risk of quinidine toxicity are under investigation [144]. The AVP-786 formulation consists of deuterated-DM and ultra-low dose quinidine. The phase II clinical trial results for TRD have yet to be reported. Phase III results (TRIAD-1/2) for Alzheimer’s-related agitation have generated conflicting results per Avanir’s/Otsuka’s press releases; however, full publications are yet to be released [153].

AXS-05/Auvelity (combination DM and bupropion) leverages bupropion’s norepinephrine–dopamine reuptake inhibitor (NDRI) and CYP2D6 inhibition to potentiate DM glutamatergic effects. It also acts as a sigma-1 receptor agonist, which has important implications for BDNF processing and other effects [154]. AXS-05 was recently approved by the FDA following positive results from phase III trials. Two phase III RCTs have evaluated AXS-05 monotherapy for MDD/TRD, showing favorable safety and tolerability [155]. For MDD, the GEMINI trial saw AXS-05 outperform the inactive placebo comparator at all time points with substantial improvements in quality of life and decreased functional impairment [156]. The STRIDE-1 trial testing AXS-05 in TRD patients began with a bupropion-only lead in period before randomizing bupropion non-responders to receive either AXS-05 or bupropion-only placebo for an additional six weeks. Compared to bupropion alone, AXS-05 demonstrated statistically significant improvements on The Montgomery–Åsberg Depression Rating Scale (MADRS) at weeks one, two, and when averaged over the entire six-week period. The combination also trended towards superior improvement at the primary endpoint (MADRS at week 6) but failed to reach statistical significance (*p* = 0.12). Notably, AXS-05 also showed statistically significant improvements in anxiety and cognitive functioning (Axsome Therapeutics, 2020). Building on these results, investigators initiated an open-label, twelve-month phase II trial for MDD patients with suicidal ideation (COMET-SI). Findings were consistent with past controlled trials, AXS-05 was associated with functional improvements and durable anti-suicidal properties, but the AD efficacy did not reach statistical significance [157]. A recent 12-month open-label trial (EVOLVE) also demonstrated significant reductions in anxiety (HAM-A) starting at one week, reaching 70% response rates at one year in patients with MDD. This will be important to replicate in larger RCTs, as co-occurring anxiety is a significant risk factor for TRD [158].

AV-101 (L-4-chlorokyurenine or 4-CI-KYN) is a prodrug converted via the kynurenine pathway to an antagonist (7-chlorokynurenic acid) at the glycine-binding site of the NMDAR NR1 subunit that has been tested for the treatment of MDD. Results from a Phase II study of AV-101 (ELEVATE) showed that the AV-101 treatment arm did not differentiate from the placebo group on the primary endpoint (MADRS at 2 weeks) [159]. An additional small (*n*  =  19) phase II crossover of AV-101 in individuals with TRD was also negative (*p*  =  0.71, d  =  0.22) [160].

### 4.3. Glycine Site NMDA Receptor Modulation

Rapastinel (GLYX-13) is a small polypeptide that acts as a partial agonist at the glycine site of NMDA receptors [161]. Preclinical evidence suggests that rapastinel may enhance dendritic complexity and long-term potentiation via bidirectional modulation of NMDA signal transduction [161,162,163]. Based on safety and efficacy data obtained in phase I and II trials [164], rapastinel is not associated with ketamine-like dissociation and was granted fast track status and later breakthrough therapy designation for MDD in 2016 [161]. However, during more rigorous phase III RCTs, IV rapastinel failed to separate from placebo as an adjunctive strategy and monotherapy (NCT02943564 and NCT03675776, respectively), and the drug’s AD development was discontinued, terminating future trials [161].

Apimostinel (NRX-1074/AGN-241751) is a higher-potency rapastinel analog with increased bioavailability. Developed for adjunctive use for TRD, apimostinel acts as a functional antagonist at the glycine B site of NMDA receptors [165]. Apimostinel has been tested in two phase I studies and a larger phase II RDBPCT, showing rapid AD effects after a single, high-dose IV administration [165]. Another phase II RCT of repeated administration oral apimostinel monotherapy is upcoming [166].

Zelquistinel (GATE-251) is an orally active, non-peptide, small-molecule NMDAR modulator. Unlike Rapastinel/Apimostinel, it acts independently of the glycine ligand site within the NMDAR complex to induce excitatory activity. This novel NMDA binding site has demonstrated antidepressant effects sustained at two weeks in chronic rodent stress models. It similarly appears to activate Akt/mTOR signaling, with subsequent elevation of synaptic proteins in the PFC. Results of a recent phase IIa clinical trial in MDD (NCT03726658) have yet to be posted.

D-cycloserine (DCS) is a broad-spectrum antibiotic, first suggested to promote antidepressant properties in 1959 by G.E. Crane [167]. At low doses (<100 mg/day), DCS acts as a functional NMDA receptor antagonist, whereas relatively higher doses (>750 mg/day) accelerate glycine site NMDA receptor potentiation and produce DCS’s antidepressant and anxiolytic effects [168]. Studies using transcranial direct current stimulation (tDCS) have shown that DCS modulates neuroplasticity through long-term potentiation via action on cortical NMDA receptors in humans [168,169]. This effect signifies therapeutic relevance for conditions (e.g., dementia) and treatments (e.g., exposure therapy) relating to learning and memory [168]. In an initial small study of TRD patients, adjunctive DCS failed to separate from placebo in terms of AD efficacy [170]. The same group of researchers later conducted a larger (*n* = 26) RCT, in which high-dose DCS augmentation showed superior AD efficacy to placebo [171]. This was replicated by another group who randomized TRD patients (*n* = 50) to receive DCS or placebo in combination with intermittent theta-bust stimulation (iTBS). Significantly greater improvements were seen with adjunctive DCS compared to placebo (d = 0.96) [172]. Though further replication is required with DCS, the model of leveraging pharmacological synaptic plasticity to enhance targeted treatments like neuromodulation provides an exciting avenue for translational research in the near future.

### 4.4. Metabotropic Glutamate Receptor Negative Modulation

Basimglurant (RO4917523/RG7090) is a potent, selective mGluR_5_ negative allosteric modulator (NAM) with a long half-life and fair oral bioavailability. Preclinical evidence indicates robust anxiolytic and antidepressant efficacy [173]. Quiroz and colleagues tested a daily adjunctive (modified release) basimglurant in a large (*n* = 333) phase IIb RCT for refractory MDD. The study did not meet its primary AD outcome measure, but the higher dose (1.5 mg compared to 0.5 mg low dose) outperformed the placebo across several secondary AD measures [174]. There was an unusually high placebo response rate in this trial. Pooled pharmacokinetic data from four phase I and one phase II studies also suggested that clearance was twofold higher in smokers and 40% higher in males [175]. Due to the negative results on the phase IIb study conducted, this drug has not been cleared by FDA to be used for refractory MDD.

### 4.5. AMPA Receptor Positive Modulation and Other Mechanisms

Riluzole is a neuroprotective agent primarily used in the treatment of amyotrophic lateral sclerosis (ALS). Pharmacologically, it modulates both GABA and glutamate systems through several cascading mechanisms. Riluzole promotes post-synaptic GABA_A_ receptor allosteric modulation and reduces glutamatergic activity via direct NMDA and kainate receptor inhibition, blockade of tetrodotoxin (TTX)-sensitive sodium channels, and simultaneous reuptake acceleration and inhibition of release [144]. Riluzole also increases glutamate–glutamine cycling, a potential explanation for in vitro observations of enhanced neuroplasticity and amelioration of excitotoxicity [176,177], as well as in vivo reductions in depressive behaviors [144,178]. According to meta-analysis findings, riluzole augmentation and monotherapy both demonstrate sustained antidepressant and anxiolytic efficacy for depressive disorders [144,179]. In TRD, AD efficacy was seen in early open-label studies but not replicated by more rigorous clinical trials [144,179], (see Table 1).

## 5. Clinical Studies with GABA Modulation in TRD

Accumulating data support the involvement of the GABAergic system in the pathophysiology and treatment of TRD [61]. A new class of robust antidepressants have evolved from the endogenous neuroactive steroid/neurosteroid (NAS) and potent GABA_A_ PAM (Positive Allosteric Modulator) allopregnanolone (ALLO) [11,24]. Evans and colleagues (2012) demonstrated that pretreatment with ALLO prevented depressive and anxious behaviors while simultaneously normalizing chronic stress-induced hippocampal neurodegeneration, HPA axis responsiveness, and BDNF expression [63]. ALLO is implicated as a potential mediator of depression based, in part, on findings of decreased ALLO levels in depressed patients that normalized with AD treatment [11]. Exogenous and endogenous administration produced sustained anxiolytic, anticonvulsant, neuroprotective, and anti-inflammatory effects, along with AD efficacy reported to last for one month—longer than any FDA-approved antidepressant available now [24,180]. ALLO’s therapeutic mechanism is mediated largely by its GABA_A_R modulation, but likely sustained via downstream GABA gene expression regulation and transcriptional changes [24,180]. However, ALLO itself is unsustainable as a long-term treatment since it tends to revert to its metabolic precursor, progesterone—a mechanistic distinction from the proceeding analogs [63].

### GABA_A_ Receptor Positive Modulation

Brexanolone (BRX/Zulresso, formerly SAGE-547) is a GABA_A_ PAM and synthetic analogue of ALLO [24] it is the first FDA-approved treatment for postpartum depression (PPD). Extended release (60 h IV infusion) BRX demonstrated rapid, durable AD efficacy during phase I, II, and III clinical trials [181,182,183]. However, a recent meta-analysis found less decisive conclusions [184]. According to Hutcherson and colleagues, only one of the two completed phase III RCTs provided evidence of remission beyond 30 days, and, in both, results were clouded by large placebo effects, narrow population samples, and other sources of bias [183,184]. BRX was generally well-tolerated, but patients experienced suicidal thoughts and behaviors associated with BRX treatment [185]. Likewise, the 2.5-day continuous infusion requirement creates some logistical hurdles for widespread use. As with Spravato, access to BRX is currently restricted to the REMS program due to excessive sedation and sudden loss of consciousness. This represents a treatment obstacle for many mothers of neonates.

Ganaxolone is an additional ALLO-like compound that demonstrates AD efficacy in early phase II trials [186,187]. However, future studies should examine daily oral maintenance doses following a short-course IV lead in to bolster real-world efficacy and feasibility in a way that BRX cannot [187].

Zuranolone (formerly SAGE-217) is another ALLO-like compound with AD efficacy in one early phase II trial [188]. A recently published phase III trial had very positive results, showing significant clinical improvements at day 3 and sustained through the whole duration of the study at day 42 [189]. Overall, clinical trial outcomes for synthetic ALLO compounds are promising as TRD therapies, but limited by small samples, short durations, and virtually identical designs. Another very recent phase III trial compared Zuranolone versus placebo in 196 patients, showing significant positive results in improving depressive symptoms on patients with PPD on days 3, 15, 28, and 45 [190]. Due to those very recent findings, Zuranolone was approved by the FDA for the treatment of PPD [191]. Larger and longer RDBPCTs are crucial to verify the safety and efficacy of these drugs, with this recent published study changing the near future related to the feasibility of this type of drug for MDD.

## 6. Conclusions

Since initial development, three different classes of antidepressant medications have been marketed: (1) Monoamine Oxidase Inhibitors (MAOIs) and Tricyclic Antidepressants (TCAs), (2) Selective Serotonin Reuptake Inhibitors (SSRIs), and (3) Serotonin–Norepinephrine Reuptake Inhibitors (SNRIs). While safety profiles have improved, a recent systematic review demonstrated that MAOIs and TCAs still retain the highest efficacy and acceptability, respectively, amongst antidepressants [192]. However, patients that suffer with TRD still do not experience significant improvement with existing therapies and also experience unwanted side effects. The development of novel therapies will provide opportunities to discover new medications that could have enhanced efficacy in treating TRD with different mechanisms of action and a more targeted approach, allowing for better symptom relief and improved treatment response rates.

Glutamate–GABA imbalance in depressive pathophysiology has emerged as an imperative to understanding depressive pathophysiology and treatment resistance. To improve future TRD research and treatment, establishing a clear and unanimous definition of treatment resistance will be highly relevant. Of the compounds reviewed here, two are FDA-approved: esketamine (Spravato) for TRD and BRX (Zulresso) for PPD. However, both have drawbacks, including the lack of independently funded research, high abuse potential, and the REMS program mandates, which are based on safety concerns. Compared to esketamine, ketamine may produce more severe psychotomimetic/dissociative side effects, but its AD efficacy and safety has been shown repeatedly in clinical trials and held up to the scrutiny of meta-analyses. Ketamine and N_2_O are both limited by a high abuse potential, and N_2_O was discontinued from development due to side effects and abuse potential. Many of the concerns regarding N_2_O come anecdotally or from recreational users and are notably absent in clinical trials. Thus, reappraisal for its applicability, especially in refractory populations, may be warranted. For instance, enhancing post-synaptic extracellular glutamate binding is essential for the acute BDNF-mediated gains in neuroplasticity attributed to ketamine treatment, but excessive pre-synaptic glutamate release or insufficient reuptake could lead to neuronal atrophy and deteriorate post-synaptic transmission [16]. Compounds with unique multimodal mechanisms, such as memantine, riluzole, and rapastinel, have each been tested in several RCTs, but results on AD efficacy are inconclusive or negative, and each compound is associated with undesirable side effects. The rapastinel analog apimostinel achieved positive AD results in a single RCT but lacks evidence of long-term efficacy. Similarly, compounds with positive phase II RCT results, ganaxolone, DCS, d-methadone, and sarcosine have all shown AD efficacy in small samples but require further validation from larger RCTs. AVP-923 showed early signs of AD efficacy but is limited by its potentially severe heart and liver side effects. Despite mixed results in phase III trials, AXS-05 shows promise based on its unique mechanism of action, anxiolytic/pro-cognitive properties, and lack of weight gain and sexual dysfunction, which are intolerable to many patients on conventional antidepressants. Zuranolone is scheduled to receive a response from FDA in August 2023 about the approval or not since the application has been granted priority by the agency [193]. Future research should focus on rigorous translational and clinical investigation to provide relief for the more than 30% of depressed patients who do respond to currently available therapies. It should also be considered to repurpose existing drugs and evaluate their use in TRD, and it may involve new routes for drug administration, with the development of AD drugs that can be administered less frequently, targeting patients that could have more difficulty in adherence because of various factors, but, at the same time, drugs and routes that engender safe, rapid, and sustained antidepressant effects. It will be important for future treatment models to consider the nuances of glutamate signaling, especially with regard to timing, duration of action, and downstream implications. Ultimately, the full potential of many glutamatergic/GABAergic compounds may be realized by emerging research on their ability to temporarily shift set points for ease of induction of neuroplasticity (i.e., “meta-plasticity”). This property may open a window during which delivery of secondary agents (or psychotherapy) can maximize and prolong beneficial adaptations [194,195].

## Figures and Tables

**Table 1 pharmaceuticals-16-01572-t001:** Efficacy Overview of Investigational Glutamatergic Compounds for the Treatment of Depression with Level of Evidence.

Compound	Sponsor	Mechanism	Side Effects	Study Source	Outcome	Sample	Design	Phase	N	Evidence Level
KETAMINE	NIMH/MayoClinic	NMDAR antagonism; AMPAR stimulation	Acute: transient dissociative and psychotomimetic effects, ⬆ HR/BP Chronic: dissociation; cognitive/locomotor deficits; renal toxicity (high abuse potential)	Berman et al., 2000	+	MDD/BPD	RDBPCT, CO, inactive placebo, IV	II	N = 7	++++
				Zarate et al., 2006a	+	TRD	RDBPCT, CO, inactive placebo, IV	II	N = 17	
				Price et al., 2009	+	TRD+SI	RDBPCT, single IV infusion [*continuation trial for responders⟶* aan het Rot et al., 2010]	II	N = 26	
				aan het Rot et al., 2010	+	TRD	Pilot OL, repeated infusion [*continuation of* Price et al., 2009 *IV ketamine-responders*]	I	N = 10	
				Mathew et al., 2010	+	TRD	Pilot RCT, single IV dose with lamotrigine pre-treatment & successive continuation riluzole	IV	N = 26	
				Diazgranados et al., 2010a	+	TRBPD	RDBPCT, CO, active placebo, IV, adjunct	II	N = 18	
				Diazgranados et al., 2010b	+	SI+TRD	Single open label IV infusion	II	N = 33	
				Zarate et al., 2012	+	TRBPD	RDBPCT, CO, inactive placebo, IV, adjunct	II	N = 15	
				Murrough et al., 2013	+	TRD	RDBPCT, PA, active control, single IV infusion	II	N = 72 (47)	
				Sos et al., 2013	+	MDD	*a priori* RDBPCT, CO, inactive placebo, IV	III	N = 27	
				Lapidus et al., 2014	+	TRMDD	RDBPCT, CO, inactive placebo, IN	N/A	N = 20	
				Hu et al., 2016	+	TRMDD+SI	RDBCT, PG, inactive placebo, single IV infusion, adjunct to new OAD escitalopram	N/A	N = 30 (15)	
				Singh et al., 2016a	+	TRD	RDBPCT, inactive placebo, repeated admin, IV	II	N = 67	
				Fava et al., 2018	+	TRD	RDBPCT, IV adjunct, active placebo, multiple doses, 5-groups (including one control group)	II	N = 99 (80)	
				Phillips et al., 2019	+	TRD	RDBCT, CO, midazolam placebo, repeated/maintenance IV infusions	III	N = 41	
				Domany et al., 2019	+	TRD/SI/HC	POC, RDBPCT, repeated oral adjunct, inactive placebo	III	N = 41 (22)	
				Ionescu et al., 2019		TRD+SI	RDBPCT, repeated dosing, IV, adjunct	N/A	N = 26 (13)	
				Wilkinson et al., 2019	+	severe/TR mood disorders	Chart review case series of repeated IV dosing titrated by weight, uncontrolled	N/A	N = 54	
				Roy et al., 2020	+	Adolescent TRD	Single group open label repeated IV dosing; with pre/post administration MRS imaging	II	N = 13	
				Wilkinson et al., 2017	+	SI	Systematic Review/Meta-Analysis of 10 controlled trials testing a single IV infusion	SR|MA	N = 167	
				Correia-Melo et al., 2020	+	TRD	RDBACT, ketamine group; esketamine group	II	N = 63	
ESKETAMINE/(*S*)–KETAMINE (SPRAVATO)	J&J (Janssen)	NMDAR antagonism (non-selective/non-competitive)	Transient dissociation (less than ketamine), sedation, anxiety, hypoesthesia, vertigo, dizziness, dysgeusia, GI disturbances, ⬆ BP	Singh et al., 2016b	+	TRD	RDBPCT, IV, adjunct, x2 randomized, PA, optional OL continuation phase	II	N = 30 (20)	+++++
				Canuso et al., 2018	+	TRMDD+SI	RDBPCT, IN, adjunct; *concluded ineffective for SI*	II	N = 68 (35)	
				Daly et al., 2018	+	TRD	RDBPCT, IN, multiple doses, adjunct, partial OL	II	N = 67 (34)	
				Ochs-Ross et al., 2018		Geriatric MDD	RDBPCT, active control, flexible dosing, IN	III	N = 138	
				Daly et al., 2019	+	TRD	Long-term RDB withdrawal study, adjunct, IN	III	N = 297	
				Fedgchin et al., 2019	+	TRD	RDBPCT, IN, active placebo, adjunct; 3 arm	III	N = 346 (233)	
				Popova et al., 2019	+	TRD	RDBPCT, active control, adjunct, flexible dosing	III	N = 227	
				Fu et al., 2020	+	MDD+active SI	RDBPCT, IN, adjunct, inactive placebo; plus optional follow up continuation (n=192)	III	N = 226 (114)	
				Ochs-Ross et al., 2020	-	Geriatric TRD	RDBACT, adjunct, IN, flexible dosing, new OAD	III	N = 138 (72)	
				Wajs et al., 2020	-	Geriatric TRD	Long-term, OL, uncontrolled, adjunct, IN [continuation of Ochs-Ross et al., 2020]	III	N = 802	
				Ionescu et al., 2020	+	MDD+active SI	RDBPCT, IN, inactive placebo, new OAD	III	N = 227 (115)	
				Correia-Melo et al., 2020	+	TRD	RDBACT, ketamine group; esketamine group	II	N = 63	
				Zaki et al., 2023	+	TRD	OL, long-term extension study	III	N = 768	
Dextromethorphan (DM) Compounds	—	NMDAR antagonism; low affinity σ_1_ opioid receptor agonism; SERT/NET inhibition	GI disturbances, dizziness, QTc prolongation (in those with heart conditions)	Lee et al., 2012 *with corrigendum*	-	BPD	RDBPCT, PA, stratified, adjunct	III	N = 309 (203)	Ø
AVP-923/Nudexta [DM+quinidine]	Avanir	NMDAR antagonism; σ_1_ agonism; SNRI	Nausea, dizziness, peripheral edema, rare liver malfunction	Messias & Everett, 2012	+	TRD	Case report on emotional lability in depression	N/A	N = 1	+
				Kelly & Lieberman, 2014	+	TRBPD-II	Preliminary retrospective chart review/case series, adjunct; high dropout rates due to nausea	N/A	N = 77	
				Murrough et al., 2017b	+	TRD	POC, repeated BID administration, OL, single arm	IIa	N = 20	
AXS-05 [DM+bupropion]	Axsome	NMDAR antagonism; σ_1_ agonism; SNRI; nACh antagonism	Anxiety, restlessness, dry mouth, arrhythmia, irritability, insomnia, hyperventilation	Iosifescu et al., 2022	+	MDE	RDBPCT, PA, active control, oral	II	N = 97	+++++
				Tabuteau et al., 2022	+	MDD	RDBPCT, PA, 2 arm, oral, repeated administration	III	N = 327 (163)	
				NCT02741791/STRIDE-1	-	TRD	RDBPCT with active placebo, parallel groups, proceding open-label bupropion lead in period	III	N = 312 (156)	
LANICEMINE (AZD6765)	AstraZeneca	low trapping NMDAR channel blocker	No dissociative or psychotomimetic effects	Zarate et al., 2013	-	TRD	RDBPCT, CO, single IV infusion, inactive placebo	II	N = 22	Ø
				Sanacora et al., 2013 (NCT00491686)	-	TRMDD	RDBPCT, single administration, IV, parallel groups, monotherapy, PILOT	IIa	N = 34 (16)	
				Sanacora et al., 2013 (NCT00781742)	+	TRMDD	RDBPCT, multiple infusions, IV, parallel groups, 3 arm, adjunctive, inactive placebo	IIb	N = 152 (102)	
				Sanacora et al., 2017	-	TRMDD	RDBPCT, PA, IV, 3 arm, inactive placebo, repeated administration, adjunct	IIb	N = 302 (202)	
RILUZOLE	Stanley Medical Research Institute/NIMH/Yale University	AMPAR stimulation; NMDAR inhibition; ⬆ VGLUT reuptake	GI disturbances, dizziness, drowsiness	Zarate et al., 2004	+	TRD	open label monotherapy	II	N = 19	+/−
				Zarate et al., 2005	+	BPD	Adjunct to lithium, OL, non-randomized	II	N = 14	
				Sanacora et al., 2007	+	TRD	Adjunct to traditional antidepressant, OL	II	N = 10	
				Brennan et al., 2010	-	BPD	OL, adjunct, brain imaging of POC/ACC, 1 arm	N/A	N = 14	
				Salardini et al., 2016	+	MDE	RDBPCT, parallel groups, adjunct to citalopram	II & III	N = 64 (32)	
				Mathew et al., 2010	-	TRD	RDBPCT, adjunct to single IV dose ketamine	IV	N = 26	
				Ibrahim et al., 2012a	-	TRD	RDBPCT, adjunct to single dose ketamine	IV	N = 42	
				Niciu et al., 2014	-	TRD	adjunct to ketamine infusion, parallel groups, oral, flexible dose, RDBPCT	IV	N = 52	
				NCT00376220	-	BPD	RDBPCT, PA, repeated administration, inactive placebo, gradual dose titration	II	N = 94 (47)	
				Park et al., 2017	-	BPD	RDBPCT, PA, monotherapy,	II	N = 19	
				NCT01204918	-	TRMDD	RDBPCT, PA, 3 arm, adjunct to SSRI/SNRI	II	N = 104 (64)	
RISLEMENDEZ (CERC-301/MK0657)	Cerecor/NIMH	NMDAR antagonism (NR2B-selective)	No significant dissociative or psychotic effects	Ibrahim et al., 2012b	-	TRD	Pilot RDBPCT, CO, oral, monotherapy	I	N = 5	Ø
				Paterson et al., 2015b	-	MDD/SI	RDBPCT, sequential parallel, 3-arm, adjunct, repeated administration, low dose; high dropouts	II	N = 137 (81)	
MEMANTINE	NIMH/Forest Laboratories	NMDAR antagonism; σ_1_ receptor agonism	Body aches, dizziness, confusion, headache, drowsiness, insomnia, constipation, agitation, hallucinations	Zarate et al., 2006b	-	MDD	RDBPCT, parallel, monotherapy, daily oral,	III	N = 32 (16)	+/−
				Muhonen et al., 2008	-	MDD+alcoholism	Naturalistic RDBPCT, PA, escitalopram placebo, adjunct	IV	N = 80 (40)	
				Anand et al., 2012	-	TRBPD	POC, RDBPCT, parallel groups, adjunct to lamotrigine, repeated administration	IV	N = 29 (14)	
				Lenze et al., 2012	-	Geriatric MDD	Pilot RDBPCT, PA, with 12 month follow up, placebo group and healthy comparator group	IV	N = 35 (17)	
				Smith et al., 2013	-	TRMDD	RDBPCT, PA, adjunct, repeated administration,	IV	N = 31 (15)	
				Lepow et al., 2017	+	MDD	Case Series/retrospective chart review of subjects from Zarate et al., 2006, 2013; Ibrahim et al., 2012a	N/A	N = 7	
				Lavretsky et al., 2020	-	Geriatric MDD	RDBPCT, adjunct to escitalopram, 12 month naturalistic follow up	IV	N = 95	
D-METHADONE (Dextromethadone/REL-1017)	Relmada	NMDAR antagonism (non-competitive, non-opioid selective)	No ketamine-like psychotomimetic effects	Fava et al., 2022	+	MDD	RDBPCT, PA, 3 arms, adjunct, repeated dosing, oral solution dissolved in cranberry juice	IIa	N = 62	+/−
				NCT04688164 (RELIANCE I)	-	MDD	RDBPCT, 2 arms, adjunctive	III	N = 232	
				NCT05081167 (RELIANCE III)	-	MDD	RDBPCT, 2 arms, monotherapy	III	N = 232	
NITROUS OXIDE (N_2_O)	Washington University School of Medicine	NMDAR, AMPAR, KAR, nACh, & 5-HT_3_ antagonism; GABA_A_ & GlyR potentiation	Well-tolerated Anxiety, nausea, headache, sedation, high abuse potential	Nagele et al., 2015	+	TRD	Pilot RDBPCT, CO, adjunct to existing treatment, one hour inhalation of 1:1 ratio of nitrous oxide-oxygen OR placebo 1:1 ratio of nitrogen(inert)-oxygen twice over two weeks	II	N = 21	++
				Yan et al., 2022	+	TRD	RDBPCT, one hour inhalation of 1:1 ratio of nitrous oxide-oxygen OR placebo 1:1 ratio of nitrogen, one time	II	N = 44	
RAPASTINEL (GLYX-13)	Allergan	NMDAR functional partial agonism (glycine site)	Dissociation	Preskorn et al., 2015	+	TRMDD	RDBPCT, PA, single administration, multiple doses, IV, inverted U dose response curve, monotherapy	II	N = 115	Ø
				NCT02943564	-	MDD	RDBPCT, PA, IV, adjunct, two doses	III	N = 658 (421)	
				RAP-MD-01, NCT03675776	terminated	MDD	RDBPCT, PA, IV, monotherapy, two doses, terminated due to futility	III	N = 50	
				RAP-MD-02, NCT02932943	-	MDD	RDBPCT, adjunct, IV	III	N = 465 (231)	
				RAP-MD-03, NCT02943577	-	MDD	RDBPCT, adjunct, IV	III	N = 429 (206)	
				RAP-MD-04, NCT02951988	-	MDD	RDBPCT, adjunct, IV, for relapse prevention, two doses, initial open label phase	III	N = 604 (402)	
APIMOSTINEL (NRX-1074/AGN241751)	Allergan	NMDAR functional partial agonism (glycine site)	No ketamine-like psychotomimetic effects	Naurex, 2015	+	MDD	RDBPCT, single administration, IV, 4 arm	IIb	N = 140	+++
D-CYCLOSERINE	NARSAD/NYS Psychiatric Institute	NMDAR functional partial agonism (glycine site)	Well-tolerated hyperexcitability, dizziness, anxiety, fatigue, GI distress	Heresco-Levy et al., 2006	-	TRD	RDBPCT, CO, adjunct, dose too low	IIb	N = 22	+++
				Heresco-Levy et al., 2013	+	TRD	RDBPCT, PA, gradual titration to high dose	II	N = 26 (13)	
				Kantrowitz et al., 2015	+	TRBPD	adjunct, maintenance therapy after single ketamine infusion, gradual titration to high dose,	IV	N = 8	
				Newport et al., 2015	+	Depression	Meta-Analysis/Systematic Review of RDBPCT (Heresco-Levy et al 2006, 2013)	MA|SR	N = 48	
				McGirr et al., 2022	+	Depression	RDBPCT, iTBS plus placebo or DCS	II	N = 50	
SARCOSINE	—	GlyT1 inhibition (⬆ NMDAR activity)	Well-tolerated	Huang et al., 2013	+	MDD	RDBPCT, citalopram control	II	N = 40	+++
BASIMGLURANT (RG-7090/RO4917523)	Hoffman-Roche/Chugai	mGluR_5_ NAM	GI disturbances, dizziness	Quiroz et al., 2016	-	TRMDD	RDBPCT, parallel group, 3-arm, adjunct, modified-release basimglurant	IIb	N = 333 (223)	+
ARKETAMINE (PCN-101)	Perception Neuroscience	NMDAR antagonism	Transient dissociation, nausea, dizziness, somnolence, numbness, blurred vision, ⬆ BP	Leal et al., 2021	+	TRD	Pilot OL, single infusion	I	N = 7	+/−
				Leal et al., 2023	-	TRD	Pilot RDBPCT, crossover study	II	N = 10	
				NCT05414422	-	TRD	RDBPCT, 3 arm	IIa	N = 102	
AV-101 (4-Chlorokynurenine/4-CI-KYN)	VistaScience	NMDAR glycine binding site	Headache, drowsiness, MSK pain, sleep disturbances	NCT03078322/ELEVATE	-	MDD	RBDPCT	II	N = 180	−
				Park et al., 2020	-	TRD	RDBPCT, crossover study	II	N = 22	
MIJ-821 (CAD9271)	Novartis	NMDAR antagonism	Amnesia, dizziness, somnolence	Ghaemi et al., 2021	+	TRD	RDBPCT, parallel group, 3-arm	II	N = 70	+++
DECOGLURANT	Roche Pharmaceutical	mGlu2/3 Receptor Antagonist	Headache, somnolence, orthostactic hypotension, nausea, dizziness, orthostatic ⬆ HR	Umbricht et al., 2020	-	TRMDD	RDBPCT, parallel group, 4-arm, adjunct	II	N = 357	Ø
TS-161 (TP0473292)	Taisho Pharmaceutical	mGlu2/3 Receptor Antagonist	Nausea, dizziness, vomit	Watanabe et al., 2021	+	TRD	RDBPCT, parallel group	I	N = 8	+/−
				NCT04821271	ongoing	TRD	RDBPCT, 2 arm, crossover study	II	N = 25	

Abbreviations: GABA = Gamma-Aminobutyric Acid; NMDAR = N-methyl-D-aspartate Acid Receptor; AMPAR = α-amino-3-hydroxy-5-methyl-4-isoxazolepropionic Acid Receptor; KAR = Kainate Receptor; mGluR = Metabotropic Glutamate Receptor; NAM = Negative Allosteric Modulator; PAM =Positive Allosteric Modulator; NAS = Neuroactive Steroid/Neurosteroid; ALLO = Allopregnanolone; PPD = Postpartum Depression; MDD = Major Depressive Disorder; BPD = Bipolar Depression; TR = Treatment Resistant; SI = Suicidal Ideation; SUD = Substance Use Disorder; SNRI = Selective Norepinephrine Reuptake Inhibitor; SERT = Serotonin Transporter; NET = Norepinephrine Transporter; VGLUT = Vesicular Glutamate Transporter; GlyT1 = Glycine Transporter 1 (expressed at glutamatergic synapses); GlyR = Glycine Receptor; nACh = Nicotinic Acetylcholine Receptor; 5-HT_3_ =Serotonin; RDBPCT = Randomized Double-Blinded Placebo-Controlled Trial; PA = Parallel Assignment; TRMDD = Refractory Major Depressive Disorder; PG = Parallel Groups Assignment; HC = Healthy Control Subjects; OL = Open Label Design; OAD = Oral Antidepressant; RDBACT = Randomized Double Blind Active Control Trial; BID = Twice Daily; IV = Intravenous Administration; IN = Intranasal Administration; CO = Crossover Study Design; PA = Parallel Arm Study Design; SR|MA = Systematic Review/Meta-analysis; POC = Proof-of-Concept Trial; NARSAD =National Alliance for Research on Schizophrenia and Depression; NIH = National Institutes of Health; NIMH = National Institute of Mental Health; POC = Parieto-Orbital Cortex; ACC = Anterior Cingulate Cortex; GI = Gastrointestinal; MSK = Musculoskeletal.

**Table 2 pharmaceuticals-16-01572-t002:** Efficacy Overview of Investigational GABAergic Compounds in the Treatment of Depression with Level of Evidence.

Compound	Sponser	Mechanism	Side Effects	Study Source	Sample	Outcome	Design	N	Phase	Evidence Level
BREXANOLONE (SAGE-547/Zulresso)	Sage	GABA_A_ NAS PAM (ALLO analog)	Sedation, acute loss of consciousness, flushed skin/face, dry mouth, vertigo	Kanes et al., 2017a	severe PPD	+	Single-arm, OL	N = 4	II	+++++
				Kanes et al., 2017b (NCT02614547)	TRPPD	+	RDBPCT parallel group	N = 21 (10)	II	
				Meltzer-Brody et al., 2018 (NCT02942004)	PPD	+	multicenter, RDBPCT, 3-arm	N = 138 (45)	III	
				Meltzer-Brody et al., 2018 (NCT02942017)	PPD	+	multicenter, RDBPCT, 2-arm	N = 108 (54)	III	
				Hutcherson et al., 2020	PPD	+/−	Review of above 3 RCTs and 1 quasi-experimental study	N = 271 (160)	SR|MR	
ZURANOLONE (SAGE-217)	Sage	GABA_A_ NAS PAM (ALLO analog)	Headache, dizziness, nausea, somnolence	Gunduz-Bruce et al., 2019	MDD	+	RDBPCT	N = 89 (45)	II	+++++
				MOUNTAIN (Sage, 2019)	MDD	-	RDBPCT, 3-arm	N = 581 (424)	III	
				Clayton et al., 2023	MDD	+	RDBPCT	N = 543	III	
				Deligiannidis et al., 2023	PPD	+	RDBPCT	N = 196	III	
GANAXOLONE (CCD-1042)	Marinus	GABA_A_ NAS PAM (ALLO analog)	Sedation, dizziness	Gutierrez-Esteinou et al., 2019	severe PPD	+	RDBPCT, 3 doses	N = 58 (30)	II	+++
				Dichtel et al., 2020	postmenopausal women w/TRD	+	pilot OL, adjunct, uncontrolled	N = 10	N/A	

Abbreviations: GABA = Gamma-Aminobutyric Acid; NMDAR = N-methyl-D-aspartate Acid Receptor; AMPAR = α-amino-3-hydroxy-5-methyl-4-isoxazolepropionic Acid Receptor; KAR = Kainate Receptor; mGluR = Metabotropic Glutamate Receptor; NAM = Negative Allosteric Modulator; PAM =Positive Allosteric Modulator; NAS = Neuroactive Steroid/Neurosteroid; ALLO = Allopregnanolone; PPD = Postpartum Depression; MDD = Major Depressive Disorder; BPD = Bipolar Depression; TR = Treatment Resistant; SI = Suicidal Ideation; SUD = Substance Use Disorder; SNRI = Selective Norepinephrine Reuptake Inhibitor; SERT = Serotonin Transporter; NET = Norepinephrine Transporter; VGLUT = Vesicular Glutamate Transporter; GlyT1 = Glycine Transporter 1 (expressed at glutamatergic synapses); GlyR = Glycine Receptor; nACh = Nicotinic Acetylcholine Receptor; 5-HT_3_ = Serotonin; RDBPCT = Randomized Double-Blinded Placebo-Controlled Trial; PA = Parallel Assignment; TRMDD = Refractory Major Depressive Disorder; PG = Parallel Groups Assignment; HC = Healthy Control Subjects; OL = Open Label Design; OAD = Oral Antidepressant; RDBACT = Randomized Double Blind Active Control Trial; BID = Twice Daily; IV = Intravenous Administration; IN = Intranasal Administration; CO = Crossover Study Design; PA = Parallel Arm Study Design; SR|MA = Systematic Review/Meta-analysis; POC = Proof-of-Concept Trial; NARSAD =National Alliance for Research on Schizophrenia and Depression; NIH = National Institutes of Health; NIMH = National Institute of Mental Health; POC = Parieto-Orbital Cortex; ACC = Anterior Cingulate Cortex; GI = Gastrointestinal; MSK = Musculoskeletal.

**Table 3 pharmaceuticals-16-01572-t003:** Efficacy Evidence Levels.

+++++	FDA-approval for depression (positive results from Phase 3 RCTs)
++++	Support from meta-analyses/systematic reviews of Phase III RDBPCTs with positive data (N > 100 per group)
+++	Positive results in Phase II and/or in RDBPCT(s), including meta-analysis/systematic reviews (N of 30–100 per group)
++	Positive results in smaller RCTs (N < 30 per group)
+	Preliminary positive results in open-label, uncontrolled, observational, OR case series
+/−	Results vary between/within studies; inconclusive efficacy based on available data
Ø	FDA rejected or overall negative/unsubstantiated results
+	Overall Positive Study Outcome
–	Overall Negative Study Outcome
+/−	Mixed Study Results

## Data Availability

Data sharing is not applicable.

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
