# Peer review of "Pharmacotherapies Targeting GABA-Glutamate Neurotransmission for Treatment-Resistant Depression"

_pharmaceuticals, 2023, doi:10.3390/ph16111572_

Round 1

Reviewer 1 Report

Comments and Suggestions for Authors

This is an interesting review which attempts provide a synthesised account of the  pharmacotherapies targeting GABA-glutamate neurotransmission for treatment-resistant depression (TRD)

The review is clearly written and the points made well. Below I have some comments for the authors to consider.

The authors must  authoritative previous literature on the glutamatergic hypothesis of depression and RAA must be cited, mostly papers by Trullas and Skolick; Witkin, etc.

The same is true for antidepressant-like effects of mGlu2/3 receptor antagonist, discovered first by Chaki and coworkers  (2004), and subsequently, decoglurant, an mGlu2/3 receptor NAM, failed to show its efficacy in a Phase 2 study(Umbricht et al, 22020). Currently  TS-161, a prodrug of an mGlu2/3 receptor antagonist TP0178894, is being tested in patients with treatment-resistant depression Chaki &Watanabe, 2023.

The paper  on the importance of the balance between GABA and glutamate in depression by Wieronska  & Pilc, Pharmacol &Ther 2019  should be mentioned.

Page 11 line 2 – the reference number 109 should be named.

Reviewer 2 Report

Comments and Suggestions for Authors

The presented review is devoted to the analysis of data on clinical studies of drugs of various structural classes that act on the glutamatergic and/or GABAergic systems and exhibit positive effects in the treatment of Treatment Resistant Depression. A fairly detailed analysis of a large set of publicly available data over the past few years has been analyzed, which is of interest to trained researchers. The main comments from the review include:
1. some illogicality in the names of the sections, namely sections 4.1 and 4.2 in the title reflect the compounds, while the names of the remaining sections reflect the mechanism of action of the compounds described below. It would be better to bring everything into line. In addition, sections 4.1.4- and below describe non-kephalin compounds, which are reflected in the title of section 4.1.
2. in section 4.2.2 the composition of the drug AVP-923 should be clarified
3. Perhaps it makes sense to include data from Table 3 in the legends of tables 1 and 2, without creating a separate table 3, which also contains duplicate designations with different meanings (bold).
4. all references and bibliography must be formatted in MDPI format

Reviewer 3 Report

Comments and Suggestions for Authors

The review presented by Vecera is comprehensive and compelling in its presentation.

For the most part I think the manuscript can be published as is. 

Minor suggestions include the addition of drug repurposing as concept in the search for therapeutic agents.

A major suggestion I have is to consider an alternate presentation or layout for the tables presented. As these are difficult to read.

In addition, the authors comment on future directions for novel research. I understand the clinical focus of the review. Yet I wonder if the authors could provide similar comments to aid molecular design endeavours.
